# Improved Bone Regeneration Using Biodegradable Polybutylene Succinate Artificial Scaffold in a Rabbit Model

**DOI:** 10.3390/jfb14010022

**Published:** 2022-12-30

**Authors:** Giulio Edoardo Vigni, Giovanni Cassata, Giusj Caldarella, Roberta Cirincione, Mariano Licciardi, Giovanni Carlo Miceli, Roberto Puleio, Lorenzo D’Itri, Roberta Lo Coco, Lawrence Camarda, Luca Cicero

**Affiliations:** 1Department DICHIRONS, Orthopaedics and Traumatology, University of Palermo, 90127 Palermo, Italy; 2Centro Mediterraneo Ricerca e Training (Ce.Me.Ri.T), Istituto Zooprofilattico Sperimentale della Sicilia “A. Mirri”, 90129 Palermo, Italy; 3Dipartimento di Scienze e Tecnologie Biologiche Chimiche e Farmaceutiche (STEBICEF), Università Degli Studi di Palermo, 90123 Palermo, Italy; 4Department ProMISE, Pathological Anatomy, University of Palermo, 90127 Palermo, Italy

**Keywords:** polybutylene succinate, microfibrillar scaffold, rabbit, bone reconstruction, bone regeneration, bone defect

## Abstract

The treatment of extensive bone loss represents a great challenge for orthopaedic and reconstructive surgery. Most of the time, those treatments consist of multiple-stage surgeries over a prolonged period, pose significant infectious risks and carry the possibility of rejection. In this study, we investigated if the use of a polybutylene succinate (PBS) micro-fibrillar scaffold may improve bone regeneration in these procedures. In an in vivo rabbit model, the healing of two calvarial bone defects was studied. One defect was left to heal spontaneously while the other was treated with a PBS scaffold. Computed tomography (CT) scans, histological and immunohistochemical analyses were performed at 4, 12 and 24 weeks. CT examination showed a significantly larger area of mineralised tissue in the treated defect. Histological examination confirmed a greater presence of active osteoblasts and mineralised tissue in the scaffold-treated defect, with no evidence of inflammatory infiltrates around it. Immunohistochemical analysis was positive for CD56 at the transition point between healthy bone and the fracture zone. This study demonstrates that the use of a PBS microfibrillar scaffold in critical bone defects on a rabbit model is a potentially effective technique to improve bone regeneration.

## 1. Introduction

In the case of extensive bone loss, regeneration or reconstruction of bone tissue represents one of the greatest challenges in orthopaedic and reconstructive surgery today.

Current surgical treatments involve the use of distraction osteogenesis techniques [1], the use of autografts [2] or allografts [3], or even replacement with prosthetic implants. These traditional approaches provide stability of the bone district and coverage of the bone defect. On the other hand, they require a long time for healing, have significant infectious risks and the possibility of rejection [4].

An attractive alternative to such traditional approaches is tissue engineering, which is used in regenerative medicine [5]. Regenerative medicine employs its resources in the search for biomaterials that can stimulate tissue growth, mimic the characteristics of the target tissue and promote healing [6]. In this context, next-generation bone graft substitutes are being developed as a support for bone tissue regeneration and as a vehicle for a biomolecular stimulus to induce bone repair [7]. For these purposes, three-dimensional scaffolds represent one of the most promising resources for improving bone tissue formation in vitro and in vivo. The necessary characteristics for the applicability of a scaffold are biocompatibility and biodegradability to ensure osteoconduction and osteoinduction capabilities [8]. In addition, the mechanical properties of the chosen biomaterial and the degree of porosity must also be appropriate [9]. 

Polybutylene succinate (PBS) scaffold has proven to be an excellent biomaterial in various fields, including osteogenesis [10], with high biocompatibility and easy processability by electrospinning technique [11,12]. It is a commercially available, biodegradable, thermoplastic aliphatic polyester synthesized by the polycondensation of succinic acid and butanediol. PBS is insoluble in water; the glass transition temperature is below 0 °C. and its melting temperature is 120 °C. It has excellent processability and proven biocompatibility making it a promising polymer for various biomedical applications [13,14]. Recently, a planar PBS microfibrillar scaffold was successfully implanted in a rat model, as a nerve regeneration tool [15]. A recent study demonstrated complete biodegradability and the absence of inflammation in high-resolution MRI investigation, showing complete reabsorption by 120 days post-implant [15]. Lastly, it can be sterilized with several techniques without compromising the scaffold’s properties.

The main aim of this study is to demonstrate, in an in vivo rabbit model, the osseointegration of an electrospun microfibrillar polybutylene succinate scaffold and its ability to increase bone regeneration as well. The present study was conducted in vivo on nine male New Zealand White rabbits. Two equal bone defects were created in each subject, one for each frontal bone. One defect was left to heal spontaneously while the other defect was treated with a PBS scaffold. CT-scan study and histological examination were performed at 4, 12 and 24 weeks after surgery in each group for evaluating the osseointegration progress.

## 2. Materials and Methods

### 2.1. Scaffold Preparation and Characterization

PBS and its copolymers are semi-crystalline thermoplastic polymers belonging to the aliphatic polyester family [16]. Characteristics and properties of PBS are high crystallisation rates, wide application range between −30 °C and 120 °C, high flexibility, strength and insulating capacity [17]. Our PBS scaffold was produced as a random fibre material by electrospinning. Briefly, a PBS (Poly(1,4-butylene succinate) extended with 1,6-diisocyanatohexane, Sigma-Aldrich, Gillingham, UK) solution (15% *w*/*v* in 1,1,1,3,3,3-hexafluoroisopropanol) was collected with a precision 10 mL syringe and placed in an NF 103 Electrospinning (MECC, Fukuoka, Japan). The flow (1.2 mL/h) passed through a PTFE tube and then into a steel flat needle (22 gauge) with a 15.5 cm gap between the needle tip and the collector. One high-voltage generator was employed with a positive voltage (+12.5 kV) to charge the steel capillary containing the polymer solution while the stainless-steel collector plate was maintained at ground voltage. The humidity was maintained in a range between 23% and 27%. The morphological characteristics of scaffolds were investigated with a scanning electron microscope (SEM) (Phenom PRO X SEM) operating at 5 kV. Each sample was deposited onto a carbon-coated steel stub, dried under vacuum (0.1 Torr), and sputter-coated with 15-nm thick gold (Sputter Coater LuxorAu, Luxor Tech, Nazareth, Belgium) prior to microscopy examination. The fibre morphology and diameters were evaluated using the ImageJ (1.52Q Wayne Rasband National Institute of Health, Bethesda, MD, USA) software.

A 3D structure of the scaffold was analyzed with a μCT scanner (Skyscan 1272, Bruker Kontich, Belgium) at a source voltage of 40 kV, a current of 250 mA, a total rotation of 180° and a rotation step of 0.3°. No filter mode was chosen for the acquisitions. The image pixel size was 2.6 μm and the scan duration was about 3 h for every sample. The scanning dataset obtained after the acquisition step consisted of images in 16-bit tiff format (3238 × 4904 pixels). The 3D reconstructions were carried out using the software NRecon (version 1.6.10.2) starting from the acquired projection images. The obtained 2D images had a colour depth of 8 bits with 265 grey levels. The whole set of raw images was displayed in a 3D space by the software CTVox.

### 2.2. Study Population

After exploring alternatives to animal model testing in bone scaffold research in the literature and taking into account the guidance given by the European Union Reference Laboratory for alternatives to animal testing (EU Reference Laboratory for alternatives to animal testing, EURL-ECVAM) on alternative methods and acceptable approaches, the animal model (rabbit) was used. Procedures involving animals were carried out in accordance with the Italian Legislative Decree N° 26/2014 and the European Directive 2010/63/EU. The animal experiment project was successfully approved by the Italian Ministry of Health with the following authorization n° 66-2022/PR dated 02/28/2022 (prot 28875.38).

The animals were housed and tested at the Istituto Zooprofilattico Sperimentale della Sicilia ‘A. Mirri’ with ministerial authorisation: 14/2015-UT.

The number of animals used for this project is reduced to the minimum compatible for the verification of the scientific objectives, compatible with the standards published in the literature that allows statistical evaluation. 

The present in vivo study was conducted on 9 male New Zealand white rabbits from the company Harlan Laboratories SRL Zona Industriale Azzida, 57 33049—San Pietro al Natisone (UD), with an average body weight of 4.85 kg (range: 3.5–6 kg). A sample size of 9 rabbits was required to obtain a 98% power and an average SFI at least 15 points better than the control group. A one-sided two sample *t*-test was calculated with a significance level (alpha) of 0.05. Animals were housed in polypropylene cage and kept in controlled temperature (22 ± 2 °C), humidity (50–55%) and light (12 h light/dark cycle). Animals had access to food and water ad libitum. Rabbits were randomly divided into 3 groups of three individuals each and allowed to acclimate for at least 2 days prior to experiments. 

### 2.3. Surgical Procedure

The experimental procedure was conducted under general anaesthesia and with the administration of analgesics and antibiotic therapy so as not to induce any pain, suffering or stress to the animal. 

Animals were induced to anaesthetic depth with inhaled isoflurane at 2% and then anaesthetised with intramuscular (i.m.) injection of Zoletil(r) (tiletamine/zolazepam; 10 mg/kg) and Domitor(r) (medetomidine hydrochloride; 0.5 mg/kg. 

The surgical procedure was performed by the same surgeon, in a sterile field after shaving and disinfecting the skin with iodine solution.

A full-thickness incision was made along the midline of the skull exposing both frontal and parietal bones. Retracted the skin flap, an approximately 8mm full-thickness circular defect was created on one frontal bone using a high-precision surgical drill (hand drill) under constant saline irrigation. An equal defect was created on the contralateral frontal bone. The diameter of the defects was checked with a ruler under microscopic vision.

All procedures were performed avoiding injury to the dura mater and underlying brain tissue. Then, the 8 mm × 2.5 mm scaffold was placed in one of the randomly chosen bone defects without exerting any pressure on the underlying tissue (Figure 1). 

At the end of the procedure, each subject had two bone defects. One defect was treated by applying the scaffold while the other was left to heal spontaneously [18]. No suturing of the periosteum is performed. The skin is sutured with 3-0 silk and disinfected with iodine solution (Betadine).

Intramuscular atipamezole (Antisedan) (300 μg/kg) was used to awaken all rabbits. Carprofen (5 mg/kg) and Enrofloxacin (5 mg/kg) were daily administered for 1 week to each rabbit. After the procedure, each animal was assigned with an identification number and housed one per cage. They were monitored on a daily basis for infection, self-mutilation, and signs of distress.

### 2.4. Imaging by CT Scanning

CT images were acquired with a Simens SOMATOM Definition AS machine, 128 banks, rotation time 0.5 s, maximum mAs 250, effective 100–120 mAs, Kilovolt (KV)120, thickness 1.2 mm with 0.6 mm interval, pitch 0.6. The filter used was B20fsmooth and window W450C40. 

The bone reconstruction was done with 0.6 mm thickness with 0.3 mm interval, B60fsharp filter, “osteo” window W1500C450.

As in the 2018 study by Pihlman H. et al. [19], the area of the defect covered by mineralised bone tissue was assessed by CT evaluation.

### 2.5. Histological Examination

After the CT examination, the subjects were euthanised with Tranax (1 mL) intracardiacally. After shaving, a median incision was made to expose the frontal bones. Using a high-precision surgical burr, the frontal bones were then harvested (Figure 2).

Specimens were fixed and preserved in 10% buffered formalin and were decalcified using Na EDTA (10% *w*/*v*, pH 7.2) prior to histological analysis. The decalcified frontal bone samples were cut transversely in two from the centre of the defect. All samples were embedded in paraffin and sectioned at a thickness of 5.0 μm for staining with haematoxylin and eosin (H&E). 

The resulting preparations were dried overnight in an oven at 37 °C. They were sparged with xylene for 20 min. After a series of passages in decreasing alcohol (100°, 95°, 75° and 50°), the slides were washed in distilled water and then stained with haematoxylin and eosin. This was followed by dehydration in ascending alcohols (50°, 75°, 95° and 100°) and clarification in xylene. After this step, the slides were mounted in acrylic mounting medium (Eukitt®, O. Kindler GmbH, Freiburg, Germany).

The stained samples were analysed for tissue infiltration, bone formation patterns and scaffold integration. Images of the stained slides were obtained with a Leica DMR microscope equipped with a Leica DFC 320 digital camera and analysed using digital image analysis (Nikon NIS Br, Nikon Instruments Europe BV, Amsterdam, The Netherlands).

An immunohistochemical evaluation of the samples was also performed for CD56 (to assess the presence of osteoblasts), CD68 (to assess the presence of osteoclasts) and CD34 (to assess the presence of vessels). In all treated and control samples, a semi-quantitative count was performed in an area corresponding to 3 contiguous high-power fields (HPF, ×400) using immunohistochemical staining for CD56 to assess osteoblast density. Based on the values obtained, the statistical average of scaffold-treated and control samples was calculated.

### 2.6. Statistical Analysis

All experiments were performed in triplicate by collecting, for each experiment, a number of samples n = 3, and values were expressed as mean ± standard deviation. The collected data were analysed within each group and for the entire population using GraphPad PrismTM 4.0 software (GraphPad Software Inc., San Diego, CA, USA). The one-way ANOVA associated with Sidak’s push-out test was used. All data were presented as mean and statistical significance was set at *p* < 0.05.

## 3. Results

### 3.1. Scaffold

Macroscopically, fiber deposition was smooth and homogeneous along the metal rod, without any gross defects (Figure 3d). An almost-linear relation between the thickness of the electrospun layer and the deposition time was observed. SEM was adopted as an investigation method to study the morphology and orientation of nanofibers. Observed under SEM (Figure 3a–b), the scaffolds appeared as a random fiber mat characterized by fiber diameters in the range between 200 nm and 1.5 µm. The scaffold has an average thickness of about 800 microns (measured by microCT, Figure 3c) and sufficient rigidity for handling during implantation (see photo in Figure 3d). There, it can be cut with scissors and adapted to the bone defect.

### 3.2. CT Study 

Bone formation was assessed by performing CT scans at 4 (Group 1, Figure 4 top), 12 (Group 2, Figure 4 center) and 24 (Group 3, Figure 4 bottom) weeks.

In the images of Figure 4, 3D reconstructions and CT scans (in a similar position) of the bone defect at 4 (top), 12 (middle) and 24 weeks (bottom) are collected. There is a magnification of the scans with the defect measurement comparing the control side with the experimental side.

The comparison in defect healing progression between the native process and scaffold application is shown in diagram of Figure 5.

The statistical analysis performed with Sidak’s test on the values measured by CT scans is described in Table 1. 

The bone regeneration that occurred from 4 to 24 weeks, from 12 to 24 weeks and in the total interval from 0 to 24 weeks was statistically significant (*p* value < 0.05) in both the control and scaffold-treated defects. No statistically significant results, due to a limited number of samples, were shown when comparing the control and scaffold group with this statistical model.

The measurements of mineralised bone tissue, expressed in percentages are shown in Table 2.

### 3.3. Histological Analysis 

The bone defect is located between the parallel bars. Comparison of samples taken at 4 and 12 weeks shows the following: the untreated defect is rich in fibrous tissue partially infiltrated by bone tissue at the periphery (Figure 6A,B). The scaffold-treated defect is occupied by new bone tissue and partially by fibrous tissue undergoing mineralization (Figure 6C,D).

The sample analysed at 24 weeks showed macroscopically two partially healed defects in both the control and the scaffold-treated defect. However, in each case there was a reduction in parietal thickness.

In the haematoxylin–eosin sections showed the following. An area of bone remodelling was evident in the untreated defect. Poorly mineralised, thinned and fragmented trabeculae with numerous areas of osteonecrosis were visible (Figure 7a). The medullary cavities showed exclusively adipose marrow.

The scaffold-treated defect, on the other hand, showed not only areas of osteonecrosis with thinned and poorly mineralised bone lamellae, but also multiple areas of osteosynthesis with evidence of immature bone tissue in formation and numerous activated osteoblasts (Figure 7b). There are also fragments of birefringent amorphous material, which may be scaffold residues (Figure 7c). We also note the presence of residual medullary cavities with clearly visible haemopoietic niches (Figure 7d). No inflammatory infiltrates of any kind are evident.

Immunohistochemical staining for CD56 (Figure 8) showed that areas of osteosynthesis were more represented in the scaffold-treated samples. The density of osteoblasts detected in these areas was also higher than in the control samples. In addition, specimens in which the lesion healed spontaneously had more areas of necrosis and fibrosis and the marker evaluation was therefore non-specific. In both tissues, control and scaffold, analysis with the markers CD34 and CD68 revealed a normal microvascular density, the percentage of haematopoietic precursor cells was <2% and the osteoclastic component was not increased. Therefore, evaluation with CD68 and CD34 produced no relevant results.

### 3.4. Complications

There were no complications involving the nine rabbits in the study. Furthermore, there were no specific complications related to the scaffold used in this study.

## 4. Discussion

The present study shows that the use of a PBS microfibrillar scaffold in critical bone defects on a rabbit model proves to be a potentially efficient technique to improve bone regeneration. 

CT examination at the last follow-up shows incomplete healing in both untreated and scaffold-treated defects. However, relevant differences are observed in the assessment of the area covered by mineralised tissue, in agreement with similar studies [18,19]. 

The statistical analysis describes bone regeneration as significant in the intervals between 0 and 24 weeks, 4 and 24 weeks, and 12 and 24 weeks in both the overall population and in the control and scaffold-treated defects. Thus, no differences, and in this case no negative effects of the application of the scaffold to the bone defect, can be detected. In the first 4 weeks, there is little appreciable healing in either defect, so the scaffold does not appear to provide an advantage. Examining the bone sections of the defect, histological evaluation reveals important information. There is no evidence of inflammatory infiltrates around the scaffold. This confirms the biocompatibility and biodegradability of the material. In 2022, Cicero et al. studied a 3D microfibrillar scaffold based on PBS (Poly 1,4-butylene succinate) produced by electrospinning in sciatic nerve lesions on a mouse model [15]. Along with an improvement in the regenerative process of the peripheral nerve, the ability to limit macrophage action and the inflammatory process counterproductive to nerve regeneration were described. Therefore, even in a different tissue type, the biocompatibility and biodegradability of PBS-based 3D scaffolds are confirmed.

Histological analysis also shows that with scaffold implantation, increased infiltration of mineralised tissue is observed. This, unlike the control, involves almost the entire surface of the scaffold. It is important to note that these results are shown to be progressive in consecutive follow-ups. Although there are no statistically significant findings of faster defect healing, certainly the histological study demonstrates a qualitative advantage. Infiltration of bone tissue, with greater amounts of mineralised tissue, contrasts with a majority of fibrous tissue. Another relevant factor in scaffold-treated defects is the presence of periosteal fibrous tissue, which is indicative of a florid osteosynthesis. Furthermore, immunohistochemical analysis, which is not indicative for CD 68 and CD34, shows instead a conspicuous positivity for CD56 at the transition point from healthy bone to the ‘fracture zone’. Immunohistochemical staining for CD56 (neural cell adhesion molecule, homophilic-binding glycoprotein with a role in cell-cell adhesion) highlights the key role of osteoblasts in bone tissue regeneration and wound healing. The presence of numerous activated osteoblasts indicates a well-started bone regeneration process and haemopoietic niches are indicative of increased bone maturation.

Although there are no statistically significant findings of faster defect healing, certainly the histological study demonstrates a qualitative advantage. The infiltration of bone tissue, with greater amounts of mineralised tissue, contrasts with a majority of fibrous tissue. Another relevant factor in scaffold-treated defects is the presence of periosteal fibrous tissue, which is indicative of a florid osteosynthesis. 

In contrast, immunohistochemical analysis shows a conspicuous positivity for CD56 at the transition point from healthy bone to the ‘fracture zone’. Immunohistochemical staining for CD56 (neural cell adhesion molecule, homophilic-binding glycoprotein with a role in cell-cell adhesion) highlights the key role of osteoblasts in bone tissue regeneration and wound healing. The presence of numerous activated osteoblasts indicates a well-started bone regeneration process and the haemopoietic niches are indicative of increased bone maturation. Immunohistochemical staining analysis for CD56, carried out to assess osteoblast density, showed that the mean osteoblast density in scaffold-treated cases is 86.5/3HPF, while the statistical average of controls is 34/3HPF; these data show that in the scaffold-treated cases osteoblast density is markedly increased compared with controls. In addition, specimens in which the lesion healed spontaneously had more areas of necrosis and fibrosis in which marker assessment was non-specific.

Immunohistochemical analysis for CD 68 and CD34 showed no relevant scaffold-associated advantages. However, the normal density of haematopoietic precursor cells and the osteoclastic component as well as the normal microvascular density show that the scaffold did not alter these processes. 

The authors consider these results to be valid evidence of the usefulness and future potential for bone regeneration of the scaffold that was the object of this study.

The pursuit of a scaffold with osteogenic, osteoconductive and/or osteoinductive properties finds considerable space in the literature, and other studies also show similar results to those described in this study. In 2012, also in a rabbit animal model with radial defect, a Chinese research group demonstrated similar properties [20]. Furthermore, it was shown how an enhanced osteogenic effect can be achieved by using an engineered scaffold with mimetic osteoinductive periosteum. Teotia et al. compared the use of 3D-printed composite porous scaffolds with resins and osteoinductive growth factors in critical bone defects on flat bone (skull) and long bone (tibia) with similar models [21]. Faster healing of scaffold-treated bone defects was described in both districts. Therefore, the present study is in agreement with these studies, considering the potential osteoconductive properties of the tested PBS scaffold. Hwan D Kim’s research group incorporated osteoinductive and osteoconductive materials, such as calcium and phosphate, into the 3D scaffolds in order to stimulate osteogenic stem cell differentiation [22]. Furthermore, it has been shown that increasing the local concentration of phosphate ions in the scaffolds promotes the activation of transcription factors, such as osteocalcin and osteopontin, which induce osteogenic differentiation in stem cells. The PBS, therefore, represents a promising biocompatible material that may have important bone regenerative properties if properly processed in a three-dimensional scaffold with a suitable fibrillar or microporous structure. The choice of producing an electrospun scaffold and thus a microfibrillar PBS-based structure, and its exact method of fabrication, represent the main focus of research in this area. The authors consider this process to be a crucial factor in the osseointegration process. In 2018, Preethi Soundarya et al. provided an overview of the different fabrication techniques for the production of scaffolds (biological macromolecules such as chitin/chitosan, collagen/gelatin, alginate, hyaluronic acid, silk, synthetic polymers, ceramics) [23]. The study, analysing more than 40 different techniques and compositions, showed interesting and positive results but concluded that an ideal manufacturing method for a scaffold has not been determined yet. Therefore, given the results of the present study, the authors believe that the microfibrillar PBS scaffold produced by electrospinning techniques may represent a viable alternative to those found in the current literature. 

The main limitation of the study is that it has a restricted number of samples, which limits the effectiveness of the statistical analysis. CT evaluation, while also used as a benchmark in other studies, has limitations due to its method of execution. A further limitation of this study may be that the histological and immunohistochemical evaluation did not include the examination of healthy bone tissue samples.

The future objective will be to engineer the scaffold in order to enhance its bone regeneration capabilities, without losing the properties demonstrated in this study.

## 5. Conclusions

The present study demonstrates that the treatment of critical bone defects with a PBS microfibrillar scaffold is an effective technique for achieving enhanced bone regeneration. The absence of adverse events demonstrates the biocompatibility and biodegradability properties of the scaffold. CT scans and histologic evaluation demonstrate qualitative and quantitative improvement in bone regeneration compared with spontaneous healing. PBS microfibrillar scaffold thus demonstrates osseointegration capabilities and its osteogenic action.

## Figures and Tables

**Figure 1 jfb-14-00022-f001:**
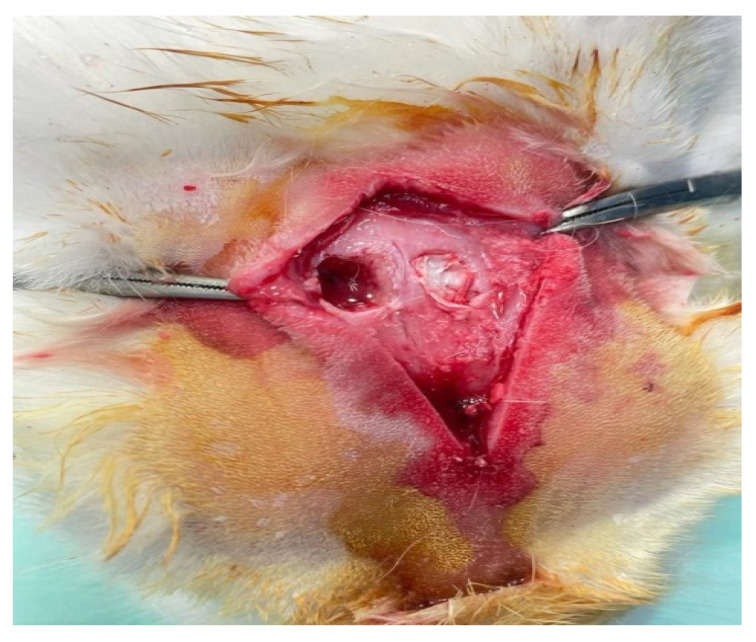
Scaffold implantation into frontal bone defect.

**Figure 2 jfb-14-00022-f002:**
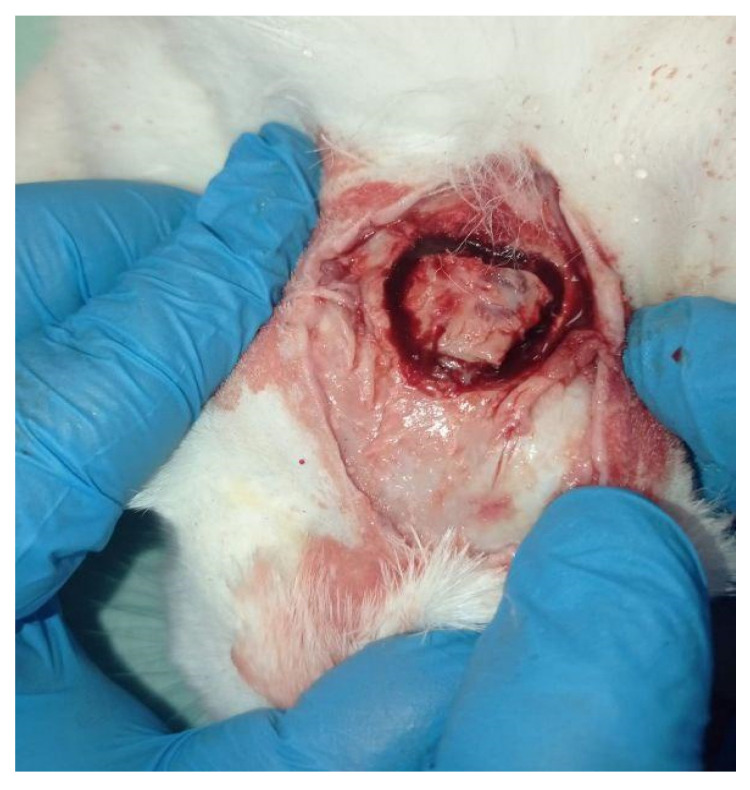
Sampling of the frontal bones.

**Figure 3 jfb-14-00022-f003:**
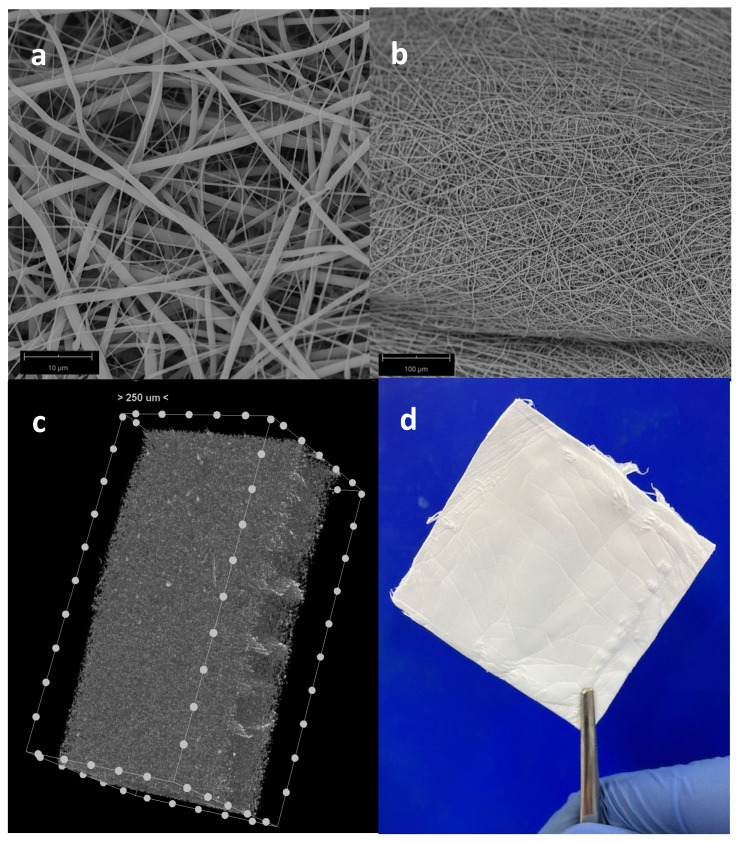
SEM image of electrospun PBS scaffold at 5000× (**a**) and 500× (**b**) magnification; microCT reconstruction of PBS scaffold (**c**) and photo of the planar scaffold (**d**).

**Figure 4 jfb-14-00022-f004:**
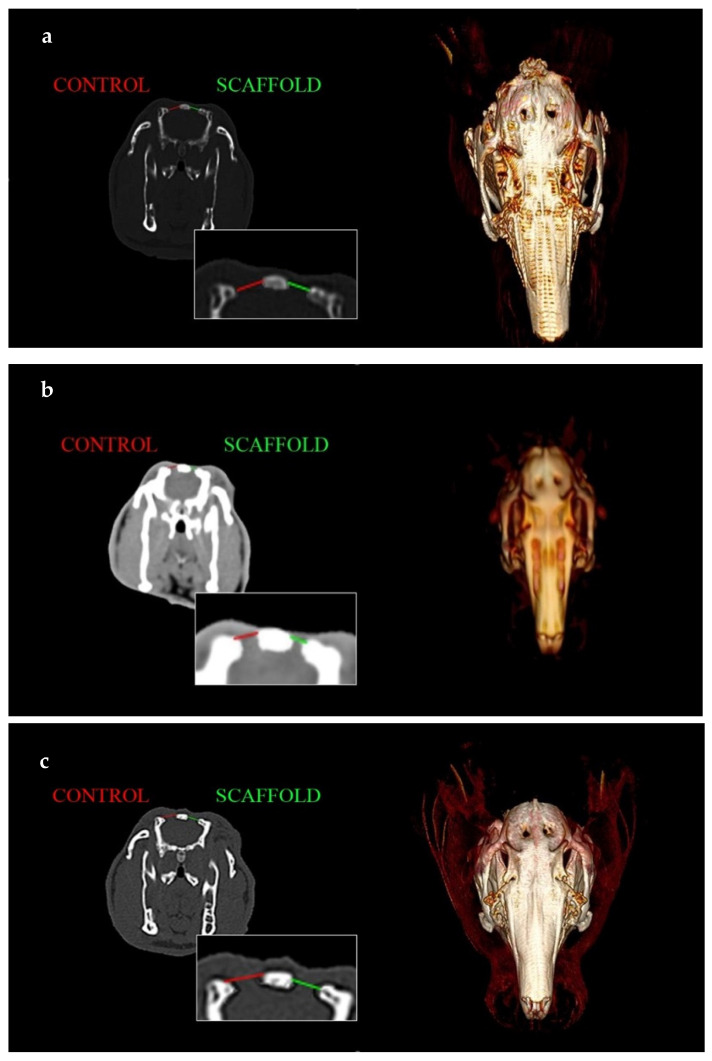
CT scan and 3D reconstruction at 4 (**a**), 12 (**b**) and 24 (**c**) weeks. Diameters’ comparison in coronal view of treated vs. untreated bone defects.

**Figure 5 jfb-14-00022-f005:**
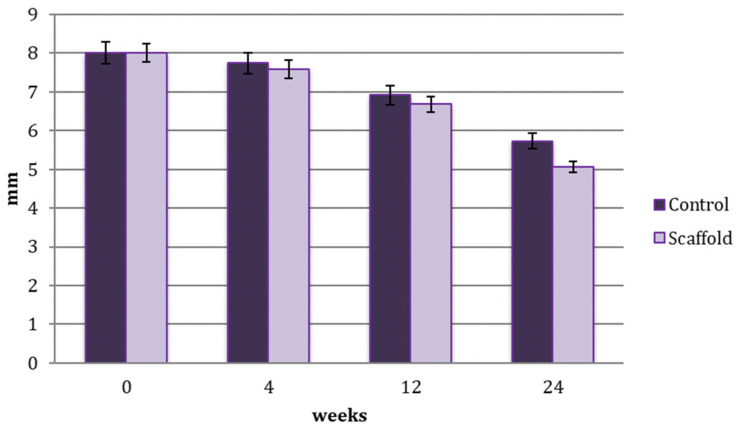
Comparison of the healing progression of the scaffold-treated and untreated (control) bone defect at 4, 12 and 24 weeks.

**Figure 6 jfb-14-00022-f006:**
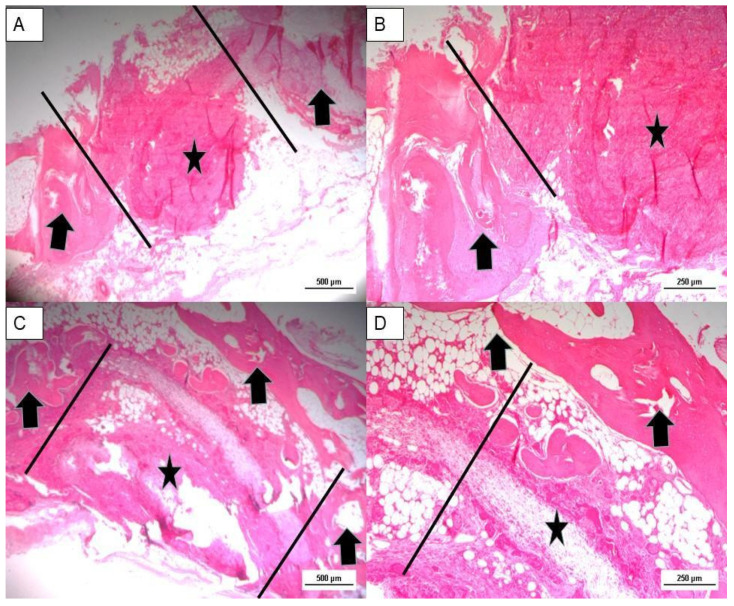
**(A**,**B**): Hematoxylin and eosin stain of untreated bone defect 4 weeks post implant (magnification: (**A**): 2.5×–(**B**): 5×). (**C**,**D**): Hematoxylin and eosin stain of treated bone defect 12 weeks post implant (magnification: (**C**): 2.5×–(**D**): 5×). Black arrows: bone tissue; black stars: fibrous tissue.

**Figure 7 jfb-14-00022-f007:**
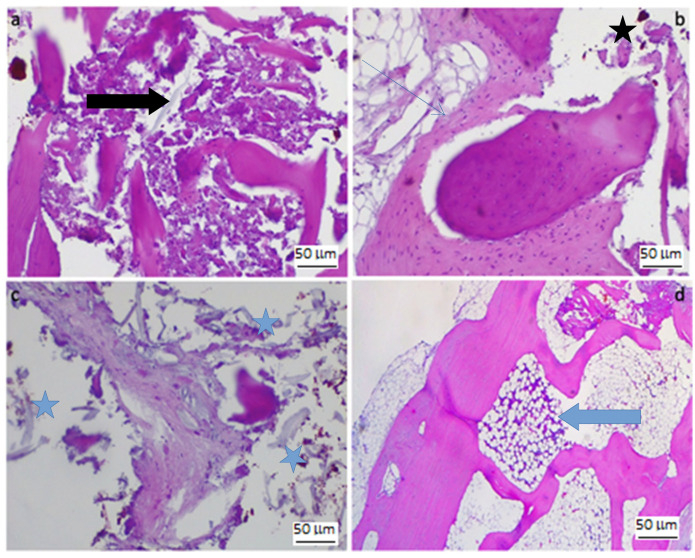
(**a**) Osteonecrosis in hematoxylin and eosin stain of treated bone defect 24 weeks post-implant (magnification: 20×); (**b**) Bone deposition and necrotic bone fragments in hematoxylin and eosin stain of treated bone defect 24 weeks post implant (magnification: 20×); (**c**) Periosseous tissue with embedded amorphous fragments (scaffold); (**d**) Bone marrow with hematopoietic stem cell niches. Black arrows: osteonecrosis areas; blue arrow: hematopoietic stem cell niches; black star: necrotic bone fragments; blue stars: amorphous fragments (scaffold); thin blue arrow: bone deposition.

**Figure 8 jfb-14-00022-f008:**
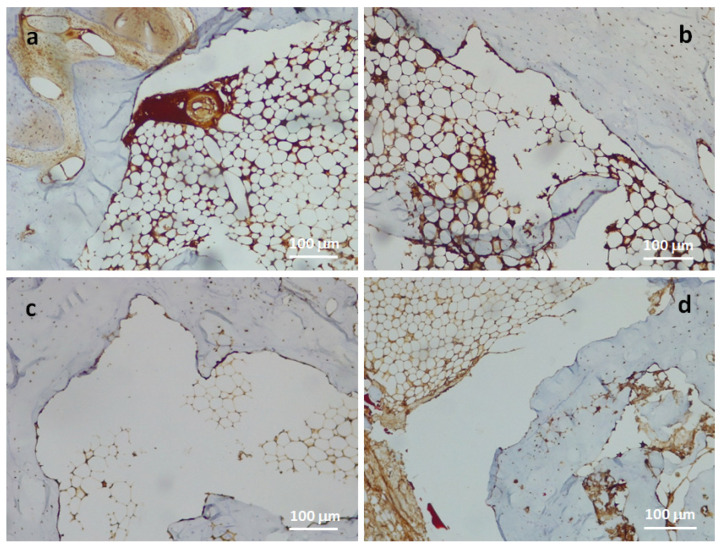
Activated osteoblasts with CD56 as immunohistochemical marker 24 weeks post-implant, related to scaffold-treated specimens (**a**,**b**) and control (**c**,**d**). Magnification 10×.

**Table 1 jfb-14-00022-t001:** Statistical Analysis on bone defects’ diameters measured using CT scans.

Sidak’s Multiple Comparisons Test	Mean Diff.	95.00% CI of Diff.	Significant	Summary	Adjusted *p* Value
0 w vs. 4 w	0.340	−0.9136 to 1.594	no	ns	0.7218
0 w vs. 12 w	1.205	−0.0486 to 2.459	no	ns	0.0558
0 w vs. 24 w	2.605	1.351 to 3.859	yes	*	0.0061
4 w vs. 12 w	0.865	−0.3886 to 2.119	no	ns	0.1336
4 w vs. 24 w	2.265	1.011 to 3.519	yes	*	0.0092
12 w vs. 24 w	1.40	0.1464 to 2.654	Yes	*	0.0367

ns, represents no significant difference. *, represent a significant difference.

**Table 2 jfb-14-00022-t002:** Percentage of defect’s surface with bone mineralization at 4, 12 and 24 weeks.

Follow-Up	Mineralised Area
4 w	~10%
12 w	~30%
24 w	~60%

## Data Availability

The data that support the findings of this study are available from the corresponding author upon reasonable request.

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
