# Peer review of "Improved Bone Regeneration Using Biodegradable Polybutylene Succinate Artificial Scaffold in a Rabbit Model"

_jfb, 2022, doi:10.3390/jfb14010022_

Round 1

Reviewer 1 Report

The manuscript conducts research on an important aspect of bone regeneration. The manuscript is well-written in most parts. I have some minor comments for further improvement.

1) The abstract states "one defect was left..." twice.

2) The statistical analysis part of the methods section needs more explanation.

3) Figure 4 needs a better description. Describe what is happening there, and possible insights.

The manuscript can be accepted after making these changes.

Author Response

Thank you for giving us the opportunity to submit a revised draft of our manuscript titled “Improved bone regeneration using biodegradable polybutylene succinate artificial scaffold in a rabbit model” to the Biomaterials for Drug Delivery section of J.F.B. We appreciate the time and effort that you all have dedicated to providing your valuable feedback on our manuscript. We are grateful to the reviewers for their insightful comments on our paper. The requested changes have been added and are highlighted in red in the text.

Here is a point-by-point response to the reviewers’ comments and concerns.

Reviewer 1:

1) The abstract states "one defect was left..." twice.

We agree with this comment. Therefore, also in accordance with another Reviewer, the abstract has been corrected.

2) The statistical analysis part of the methods section needs more explanation.

As suggested by the reviewer, the statistical analysis section was implemented and the changes are highlighted in the text.

3) Figure 4 needs a better description. Describe what is happening there, and possible insights.

We have, accordingly, improved the description of the Figure 4 in the text.

Reviewer 2 Report

Dear Authors, after a careful reading of your manuscript entitled "Improved bone regeneration using biodegradable polybutyl-2ene succinate artificial scaffold in a rabbit model" I have to announce several critical points which I summarize below and which I noted directly on the manuscript.

1) The results relating to the CT are very scarce, the type of measurements carried out for the production of the data in table 1 is not well defined (inserted in the histology paragraph instead of in that of the CT).

2) The stains in figure 5 and 6 are substantially different even if they should both be Hematoxylin and eosin.

3) Immunohistochemistry lacks control for comparative evaluation. What do you mean by nonsignificant evaluations when referring to immunohistochemistry for CD34 and CD68? Have analyzes been carried out that are not shown? What do you consider "significant"?

4) The statistical analysis relating to graph 8 describes only the differences between the times (0, 4, 12, 24 weeks) but not those between control and treated at each experimental point. Since a two way-ANOVA analysis was performed, what is the 2nd parameter considered? Is there a difference between controls and treated? This aspect is not clear by reading the results and the discussions.

5) In your manuscript you assert that your scaffold is biodegradable, where can this property be deduced from? What is the experimental evidence to support this thesis?

6) In line 327 he begins an enumeration of other works that tell of osteoinductive properties related to this material and asserts, in line 329, that your study is also in line with those mentioned. Does this mean that your material is also osteoinductive? If so, I would like to understand what the supporting data are. If not, how is your study in line with those mentioned?

7) The discussion in general looks like an enumeration of work by other research groups that are in line with your studies. The discussion should highlight the potential of your material which you claim has limited effect (line 343).

8) Also in the conclusions it is asserted that the treatment of lesions with PBS scaffolds accelerates regeneration but in the results and in the discussion no reference is ever made to significant differences between control and treated as they have always been described in histological images from which no a quantitative but only qualitative analysis.

Personally I recommend a careful new drafting of the work.

Best regards

Author Response

Thank you for giving us the opportunity to submit a revised draft of our manuscript titled “Improved bone regeneration using biodegradable polybutylene succinate artificial scaffold in a rabbit model” to the Biomaterials for Drug Delivery section of J.F.B. We appreciate the time and effort that you all have dedicated to providing your valuable feedback on our manuscript. We are grateful to the reviewers for their insightful comments on our paper. The requested changes have been added and are highlighted in red in the text.

Here is a point-by-point response to the reviewers’ comments and concerns.

Reviewer 2:

1) The results relating to the CT are very scarce, the type of measurements carried out for the production of the data in table 1 is not well defined (inserted in the histology paragraph instead of in that of the CT).

We thank the reviewer for this comment. Therefore, the text has been corrected. The CT data have all been collected in the 'CT study' section, including Tables 1 and 2.

2) The stains in figure 5 and 6 are substantially different even if they should both be Hematoxylin and eosin.

After editing the manuscript, the figures in question are number 6 and 7. We agree with the reviewer. In both cases, the samples are stained with haematoxylin-eosin. The different colouring is probably due to the fact that the sections were acquired with two different cameras. It is therefore possible that the colour saturation of the figures appears different. We believe that the slightly different colouring does not alter the understanding and significance of the figures. 

3) Immunohistochemistry lacks control for comparative evaluation. What do you mean by nonsignificant evaluations when referring to immunohistochemistry for CD34 and CD68? Have analyzes been carried out that are not shown? What do you consider "significant"?

We thank the reviewer for this comment. The manuscript has been revised so that the part in question is clearer and more precise. Specifically, sentences were corrected and added to lines 306 to 313 in the Results section and lines 347 to 381 in the Discussion section.

4) The statistical analysis relating to graph 8 describes only the differences between the times (0, 4, 12, 24 weeks) but not those between control and treated at each experimental point. Since a two way-ANOVA analysis was performed, what is the 2nd parameter considered? Is there a difference between controls and treated? This aspect is not clear by reading the results and the discussions.

We agree with the reviewer and apologise for the error. The appropriate corrections have been made. (lines 207 to 212 and lines 260 to 262)

5) In your manuscript you assert that your scaffold is biodegradable, where can this property be deduced from? What is the experimental evidence to support this thesis?

The authors are pleased to respond to this observation. Cicero et al. used PBS as a planar microfibrillar scaffold implanted as a conduit in a rat model to preserve nerve continuity and promote nerve regeneration. They observed biodegradability through multiple follow-ups and a high-resolution magnetic resonance imaging investigation that showed complete resorption in 120 days after implantation. doi:10.1002/jbm.b.34896. (lines 46 to 50).

6) In line 327 he begins an enumeration of other works that tell of osteoinductive properties related to this material and asserts, in line 329, that your study is also in line with those mentioned. Does this mean that your material is also osteoinductive? If so, I would like to understand what the supporting data are. If not, how is your study in line with those mentioned?

We thank you for your comment. Considering the immunohistochemical and CT results, we believe that the results obtained, although preliminary, confirm the potential osteoinductive properties of PBS when properly processed. The novelty of the study presented concerns the use of the electrospun scaffold as a support for bone regeneration. There is no evidence in the literature of PBS-based electrospun scaffolds used for this purpose. Therefore, we believe that a comparison with other similar studies can support and confirm the validity of the results of the present study.

7) The discussion in general looks like an enumeration of work by other research groups that are in line with your studies. The discussion should highlight the potential of your material which you claim has limited effect (line 343).

The authors thank the reviewer for this comment. As there are no identical studies in the literature, the presence of similar research investigating the same material increases the validity of our study. Now that the manuscript has been revised, we believe that the potential of our scaffold can be better understood. This is especially considering the fact that this material will have to be engineered with specific molecules.

8) Also in the conclusions it is asserted that the treatment of lesions with PBS scaffolds accelerates regeneration but in the results and in the discussion no reference is ever made to significant differences between control and treated as they have always been described in histological images from which no a quantitative but only qualitative analysis.

We thank the reviewer for this comment. The authors believe that the results obtained absolutely demonstrate the existence of a qualitative advantage confirmed by histology. Although no statistically significant confirmation was obtained, Figure 5 shows an even quantitative advantage of the scaffold. Potentially, over a larger number of treated subjects, a statistically significant advantage could have been observed between control and treated subjects. However, many changes have been made to the manuscript and we believe these differences are now easier to understand.

Reviewer 3 Report

Dear Sir-

Author Response

Thank you for giving us the opportunity to submit a revised draft of our manuscript titled “Improved bone regeneration using biodegradable polybutylene succinate artificial scaffold in a rabbit model” to the Biomaterials for Drug Delivery section of J.F.B. We appreciate the time and effort that you all have dedicated to providing your valuable feedback on our manuscript. We are grateful to the reviewers for their insightful comments on our paper. The requested changes have been added and are highlighted in red in the text.

Here is a point-by-point response to the reviewers’ comments and concerns.

Reviewer 3:

Q.1) Why author choose polybutylene succinate for the review not discussed convincingly in the introduction section? Thus, it is expected that the author should discuss the properties of polybutylene succinate in the introduction.

It is recommended to refer to and cite more research papers related to polybutylene succinate.

We have, accordingly, emphasized this point, implementing the introduction of the manuscript. (lines 53 to 62).

Q.2) Abstract is not written properly. Obtained result summary should be reflected in the abstract. But, here only the experimental part is explained in the abstract.

We agree with this comment. The abstract was rewritten including a short summary of the results.

Q.3) Section 2.1 “Scaffold preparation and characterization” fourth line Our PBS scaffold was produced following a new procedure and should be checked thoroughly. The surgical section needs to be more precise and more appropriate. 

Thank you for this suggestion, which is in agreement with the requests of the other reviewers. Detailed information about PBS and its processing for the production of the scaffold has been added to the text.

In the surgical section (2.3), corrections have been made and a clarification added. However, the surgical procedure itself is absolutely simple.

Q.4) Are Figure 1 and Figure 2 taken by the author? If “YES” then label the diagram properly.

Figures 1 and 2 were taken by the authors during surgical experiments. The figures are labelled in the manuscript.

Q.5) Remove the instrument-generated information from the SEM images.

We have, accordingly, removed the instrument-generated information from the SEM images.

Q.6) If possible provide CT scan and 3D reconstruction with high magnification and it must be labeled “a” “b” “c” and d.

Thank you for this suggestion. The recommended changes have been implemented in the text.

Q.7) The primary role of osteoblasts is to lay down new bone during skeletal development and remodeling. How to indicate the osteoblasts directly interact with other cell types within the bone, including osteocytes and haematopoietic stem cells.

We thank the reviewer for his interest in this aspect of our study. The evaluation of the interaction between various cell populations is a very interesting and complex aspect to analyse. Besides basic knowledge, however, we feel that this topic is very broad and beyond the scope of our article. Therefore, this aspect has not been explored in depth.

For example, the literature reports the presence of bone marrow stromal cells involved in mediating osteogenesis. These were present in areas with higher numbers of osteoblasts and immature osteocytes, which represented a subpopulation of multipotent cells 'involved' in bone tissue apposition.

Sources:

Ilas, D.C., Baboolal, T.G., Churchman, S.M. et al. The osteogenic commitment of CD271+CD56+ bone marrow stromal cells (BMSCs) in osteoarthritic femoral head bone. Sci Rep 10, 11145 (2020). https://doi.org/10.1038/s41598-020-67998-0

Q.8) Justification is needed why Immunohistochemical analysis shows a conspicuous positivity for CD56 in the transition zone from healthy bone to the fracture zone.

We thank the reviewer for this comment. The manuscript has been extensively rewritten in the immunohistochemistry section. We believe the topic is now clearer and more precise in both the results and discussion sections.

In addition to all previous comments, we have made minor corrections in grammar, punctuation, and citation number.

Round 2

Reviewer 2 Report

Dear authors, I regret to note that you have not re-read the paper carefully and I would say that you have not looked at the corrections I had made on the manuscript that I uploaded in the first revision. I am sending you the manuscript with the corrections. There are repetitions of entire sentences and some small typos that I have indicated (highlighted in green), moreover, you should specify, in the histological photos, arrows and asterisks.

Regarding the osteoinductivity of your scaffold, you need to perform other tests in order to demonstrate the presence of mesenchymal cells or osteoblast precursors. It would be better for you to talk about osteoconductive but not osteoinductive properties if you are not going to test other markers.

Furthermore, on lines 375-376 it is written "the total number of osteoblasts was normal"; if a quantitative analysis has not been carried out, you cannot speak of the total number of osteoblasts, but above all what would be the "normal" condition you are talking about? I suggest carrying out an analysis for example via ImageJ in which, on several photos, you carry out a count of the area covered by positivity or non-positivity at the chosen marker in relation to the total area. It is a semi-quantitative analysis that can be used as a preliminary data.

You should also insert the scale bars in the new images (fig. 8).

Best Regards

Author Response

Thank you for giving us the opportunity to submit a second revised draft of our manuscript titled “Improved bone regeneration using biodegradable polybutylene succinate artificial scaffold in a rabbit model” to the Biomaterials for Drug Delivery section of J.F.B. The requested changes have been added and are highlighted in red in the text.

Here is a point-by-point response to the reviewers’ comments and concerns.

Dear authors, I regret to note that you have not re-read the paper carefully and I would say that you have not looked at the corrections I had made on the manuscript that I uploaded in the first revision. I am sending you the manuscript with the corrections. There are repetitions of entire sentences and some small typos that I have indicated (highlighted in green), moreover, you should specify, in the histological photos, arrows and asterisks.

Dear reviewer we apologize for the fact that we have not downloaded the manuscript containing your corrections in the first revision. We thank you for the effort that you have dedicated to providing your valuable feedback on our manuscript. Now the requested changes you highlighted in green have been made.

In the histological photos, arrows and asterisks have been specified.

Regarding the osteoinductivity of your scaffold, you need to perform other tests in order to demonstrate the presence of mesenchymal cells or osteoblast precursors. It would be better for you to talk about osteoconductive but not osteoinductive properties if you are not going to test other markers.

We thank the reviewer for this comment. Therefore, accordingly the word “osteoinductive” has been corrected in “osteoconductive”in the Discussion section.

Furthermore, on lines 375-376 it is written "the total number of osteoblasts was normal"; if a quantitative analysis has not been carried out, you cannot speak of the total number of osteoblasts, but above all what would be the "normal" condition you are talking about? I suggest carrying out an analysis for example via ImageJ in which, on several photos, you carry out a count of the area covered by positivity or non-positivity at the chosen marker in relation to the total area. It is a semi-quantitative analysis that can be used as a preliminary data.

 Authors thank the reviewer for his suggestion. In order to clarify this point a semi-quantitative count was performed in an area corresponding to 3 contiguous high power fields (HPF, x400) using immunohistochemical staining for CD56 to assess osteoblast density in the scaffold-treated samples and controls.

Accordingly the following new text was added in Materials and Methods section: “In all treated and control samples, a semi-quantitative count was performed in an area corresponding to 3 contiguous high power fields (HPF, x400) using immunohistochemical staining for CD56 to assess osteoblast density. Based on the values obtained, the statistical average of scaffold-treated and control samples was calculated.”

Moreover, the sentence in Discussion section (lines 373-376) was changed with the following: ”Immunohistochemical staining analysis for CD56, carried out to assess osteoblast density, showed that the mean osteoblast density in scaffold-treated cases is 86.5/3HPF, while the statistical average of controls is 34/3HPF; these data show that in scaffold-treated cases osteoblast density is markedly increased compared with controls.”

In addition to all previous comments, we have made corrections in the figures: for figure 6 stars and arrows were defined in the legend;

for figure 7 ostonecrosis areas, fragments and niches have been indicated; for figure 8 scale bar was reported.

Reviewer 3 Report

Overall, the paper is well-orgainzed and  now the detailed discussion has made it easy to be understood.

The paper is accepted.

Author Response

Overall, the paper is well-organized and now the detailed discussion has made it easy to be understood.

 The paper is accepted.

Authors thank the reviewer for his positive comment on the revision and acceptance of the manuscript.

Round 3

Reviewer 2 Report

Dear Authors,

in the light of your latest corrections, your manuscript now seems to me to be improved compared to the original version. It is a work that could have had much more resonance if it had been presented in a more precise way.

Sincerely